# Nonlinear Analytical Procedure for Predicting Debonding of Laminate from Substrate Subjected to Monotonic or Cyclic Load

**DOI:** 10.3390/ma15238690

**Published:** 2022-12-06

**Authors:** Marco Lamberti, Francesco Ascione, Annalisa Napoli, Ghani Razaqpur, Roberto Realfonzo

**Affiliations:** 1ENEA, Research Center Brasimone, 40032 Camugnano, Italy; 2CNRS, Centrale Marseille, LMA, Aix-Marseille University, 13453 Marseille, France; 3Department of Civil Engineering, University of Salerno, 84084 Fisciano, Italy; 4Sino-Canada Joint R&D Center on Water and Environmental Safety, College of Environmental Science and Engineering, Nankai University, Tianjin 300071, China; 5Department of Civil Engineering, McMaster University, Hamilton, ON L8S 4L8, Canada

**Keywords:** SRP, semi-analytic model, cyclic loading

## Abstract

The bonding of steel/fiber-reinforced polymer (SRP/FRP) laminate strips to concrete/masonry elements has been found to be an effective and efficient technology for improving the elements’ strength and stiffness. However, premature laminate–substrate debonding is commonly observed in laboratory tests, which prevents the laminate from reaching its ultimate strength, and this creates uncertainty with respect to the level of strengthening that can be achieved. Therefore, for the safe and effective application of this technology, a close estimate of the debonding load is necessary. Towards this end, in this paper, a new, relatively simple, semi-analytic model is presented to determine the debonding load and the laminate stress and deformation, as well as the interfacial slip, for concrete substrates bonded to SRP/FRP and subjected to monotonic or cyclic loading. In the model, a bond-slip law with a linearly softening branch is combined with an elasto-plastic stress-strain relationship for SRP. The model results are compared with available experimental data from single-lap shear tests, with good agreement between them.

## 1. Introduction

After decades of service, concrete structures may exhibit insufficient strength or stiffness due to either environmental degradation or changes in loading requirements, thus creating the need for their repair/rehabilitation [1,2,3]. Relatively recently, the repair and retrofit of deficient concrete members by C (SRP) textiles or laminates have been found structurally and economically feasible. Such textiles are made of high-tensile-strength steel micro wires, twisted into small diameter cords or strands, which are unidirectionally laid in a polymer matrix to form a composite fabric. Similar to the more familiar fiber-reinforced polymer (FRP) laminates, the SRP can be externally bonded to a substrate via wet lay-up, using either epoxy or polyester resin as adhesive. The tensile behavior of steel textiles has been investigated in a number of studies, either specifically devoted to their mechanical characterization or in the context of assessing their suitability as external reinforcement for structural elements [4,5]. 

As demonstrated in the above investigations, differently from similar FRP strengthened members, SRP-retrofitted elements can exhibit relatively ductile behavior and higher energy dissipation at failure. However, the bond between the laminate and the concrete substrate is a crucial factor that affects the efficacy of this retrofit method. Consequently, analytical/numerical assessment of the retrofit requires, among other things, a suitable constitutive law for the laminate–substrate interface. 

Modeling the global behavior of existing members strengthened in bending by means of bonded FRP laminates is a major topic of interest in practice and it has been extensively researched [6,7,8,9,10,11,12,13,14], but unlike FRP, which behaves linear-elastically until failure, SRP is an elasto-plastic material with strain-hardening characteristics. Consequently, the analysis of SRP retrofitted elements can be relatively more complex than that of FRP-retrofitted elements. To date, some mainly experimental studies have been conducted to investigate the physical and mechanical properties of SRP and the behavior of SRP-retrofitted concrete elements via single-lap shear tests. Furthermore, some analytical/numerical studies involving fracture mechanics and finite element analyses have been conducted with the aim of developing some empirical models [6], following the models intended for FRP-retrofitted elements as described below. 

Teng et al. [7] studied the behavior of FRP–concrete interface between two adjacent cracks, assuming all forces applied at the two ends of the FRP strengthening plate and concrete member to remain proportional during the entire loading process. Although their study was aimed at describing the behavior of retrofitted concrete members, they applied a simplified relationship to describe the complex interfacial shear stress-slip relationship. Similarly, Chen et al. [8] developed a simple analytical solution for determining the strength of FRP–concrete bonded joints based on a linear softening bond-slip curve without allowing any slip before the maximum shear at the interface reaching or exceeding the interfacial shear resistance. Another detailed treatment of this problem was undertaken by Quiao and Chen [9], who presented a solution using a linear law to describe the bond-slip behavior of the interface. 

To characterize the debonding mechanism, numerous other models have been proposed based on linear or nonlinear fracture mechanics, regression analysis and some semi-empirical methods. The available models focus on the analysis of the stress transfer and fracture propagation in different kinds of adhesive joints by adopting different shear stress-slip models, with or without consideration of a softening branch. For example, cohesive debonding of a bonded strip from an elastic substrate has been studied by Franco. and Royer Carfagni [10] using a model representing a finite stiffener bonded to the boundary of a semi-infinite plate to determine the effective bond length of the stiffener. Cornetti and Carpinteri [11] obtained an analytical solution by assuming an exponential decaying softening branch for the interfacial bond-slip law. Accordingly, expressions for the interfacial shear stress distribution and the load–displacement response were derived for the different loading levels. In [12] the effect of different bonded lengths on FRP–concrete interfacial debonding behavior was investigated. Finally, Liu et al. [13] presented an analytical model to simulate the debonding process at the FRP–concrete interface in a single-lap shear test, but the accuracy of the model was gauged by only comparing its results with the corresponding finite element analysis results.

Using a suitable bond-slip law or relationship, analytical solutions for determining the tensile stress and strain in the FRP laminate, the failure load of the retrofitted assemblage, the shear stress distribution and the associated slip along the FRP–concrete interface, and the degree of interfacial damage due to a certain applied load can be obtained. The bond-slip relationship depends on the material properties of the adherents and the geometrical dimensions of the elements involved [14]. Since SRP as an elasto-plastic material has different properties than linear elastic FRP, the analytical models developed for FRP–concrete interfaces may not directly apply to SRP–concrete interfaces. Consequently, there is a need for simple analytical models that can be used to predict the behavior and strength of SRP–concrete interfaces. 

Another important factor that may govern interfacial failure or debonding is the type of load to which the FRP is subjected. To date, the majority of studies have focused on monotonic loading while some structures are retrofitted to increase their seismic resistance or, in the case of bridges, to increase their live load capacity. In both cases, the interface will be subjected to cyclic loading. Consequently, it is necessary to investigate the FRP/SRP–concrete interfacial behavior and ultimate strength under cyclic loads. In particular, the effect of those cyclic loadings that induce very high bond stresses must be determined to avoid premature debonding.

Some researchers have investigated the fatigue performance of the FRP–concrete interface [15,16,17,18,19,20,21,22,23,24,25,26,27,28,29,30,31]. Other studies [15,16] have shown that debonding failure due to fatigue loads is often smaller than the ultimate capacity under monotonic loading. Furthermore, the combined effects of cyclic loading and hygrothermal environment accelerate the degradation process of the FRP-to-concrete bonded interface and thus reduce the fatigue life of the interface [17,18]

Despite numerous studies dealing with the fatigue behavior of strengthened structures [19,20] and the advantages of the FRP strengthening systems to delay the crack propagation and limiting the crack width, there are still some unresolved issues with regard to the fatigue performance of the FRP-to-concrete bond [21,22,23,24,25,26], partly due to the difficulties of conducting sufficiently robust tests to comprehensively capture the fatigue behavior of the FRP/SRP-concrete interface under direct shear and ensuring the pure shear stress state of the bonded interface [22,29]. Among the various bond tests available in the literature, single and double shear tests are probably the most common. 

Zheng et al. [18] performed double shear test on specimens with carbon fiber laminate by subjecting them to fatigue load under a series of constant temperature and relative humidity (RH) conditions. The results showed that the temperature and RH negatively affected the bond behavior of the carbon–concrete interface. Other experimental double-shear tests using an improved test set-up were performed by Yun et al. [21], demonstrating the effect of different anchoring systems on the fatigue behavior of the FRP-to-concrete bond; the influence of the load amplitude and number of cycles on the bond performance. The test results showed the better fatigue performance of the hybrid-bonded FRP (HB-FRP) system compared to the other FRP strengthening systems that were examined. 

Ferrier et al. [22] studied the fatigue behavior of the bond interface using a standard double-lap shear test. The experimental results suggested a linear relationship between the maximum strength of the interface and the logarithm of the number of load cycles. Particulary, Mazzotti et al. [26] presented the results of an experimental program involving the cyclic behavior of FRP–concrete interface while Nigro et al. [27] reported the results of fatigue tests on single-lap shear specimens. Ko et al. [29] proposed a bond-slip model intended to simulate the observed response of a series of specimens retrofitted with aramid, carbon or polyacetal FRP strips bonded to concrete blocks and subjected to monotonic or cyclic load. In the model a Popovics-like constitutive law was assumed and it involved seven mechanical parameters that must be calibrated experimentally. Finally, Martinelli et al. [30], formulated a model using two different expressions to represent the bond-slip behavior of FRP strips bonded to concrete substrate. 

Considering the need for a robust, yet simple, analytical model that can be used to analyze the interfacial behavior and obtain the ultimate capacity of FRP- (Carbon, glass, aramid, etc.) or SRP-retrofitted concrete elements, in the current investigation such a model is proposed. The proposed model is intended for application to elements subjected to monotonic or cyclic load. Essentially, a closed-form analytical solution is presented to rapidly analyze the interfacial response of linear elastic or elasto-plastic laminates bonded to concrete or other similar substrates subjected to monotonic or cyclic load. The accuracy and robustness of the model are demonstrated by analyzing many test specimens and comparing the analytical results with companion test results reported by others. Besides the ultimate capacity of the retrofitted element, the model can provide interfacial shear stress distribution as well as the laminate stress and strain along its bonded length. 

## 2. Governing Equations

The purpose of the present investigation is to provide an analytical model that can be used to study the pull-out behavior and interfacial debonding of a FRP or SRP laminate bonded to a concrete substrate as encountered in single lap shear tests. The geometry of a typical single-lap shear test is represented in Figure 1, where the symbol L_f_ denotes the bonded length, bf the width of the reinforcing strip, and b_c_ and h_c_ the width and the height of the concrete prism, respectively. The pull-out specimen illustrated in Figure 1 is composed of a FRP or steel strip and a concrete prism, which can be treated as two adherents subjected to axial deformation only, while the adhesive layer that bonds the two can be assumed to be subjected only to shear deformation. Bending effects are neglected, while the shear stress and the axial deformation are assumed to be constant across and through the thickness of the adhesive layer and across the width of the laminate. 

To derive the basic equations of the analytical model, the equilibrium of the forces acting on an infinitesimal element of length *dx* along the strengthened prism is shown in Figure 2.

Based on the horizontal equilibrium of the forces in Figure 2,
(1)$$\frac{dN_{f}}{dx}-\tau b_{f}=0$$
where $\;N_{f}=\sigma_{f}b_{f}t_{f}\;$ is the axial force resisted by the strengthening strip or laminate, $\sigma_{f}$ the axial stress of strip or laminate, τ(x) the shear stress of adhesive layer.

The interfacial slip is the difference between the horizontal displacement of FRP laminates, $u_{f}$ and the concrete surface, $u_{c}$:(2)$$\lambda =u_{f}-u_{c}$$

The evaluation of $u_{c}$ requires the introduction of the equation of equilibrium of the axial forces acting on the specimen cross section as follows
(3)$$N_{c}+N_{f}=\sigma_{c}h_{c}b_{c}+\sigma_{f}t_{f}b_{f}=0$$

It is important to point out that Equation (3) allows consideration not only of the geometry of the concrete prism section, but also its mechanical properties without disregarding substrate axial strain.

### The SRP-Concrete Interface and the Pertinent Materials Constitutive Laws

The tensile behavior of steel textiles has been experimentally investigated by De Santis et al. [4,5] through traditional tensile tests, some of whose results are recapped in Figure 3. In this figure, it can be observed that the tensile stress-strain relationship of SRP is initially linear elastic up to 60–70% of its tensile strength, followed by a nonlinear segment that exhibits gradual stiffness reduction until the specimen rupture. This nonlinear pre-rupture behaviour is due to the intrinsic ductility of the steel cords and the partial unwinding of the twisted wires that form the cords. Cords with relatively steep twist angles are reported to exhibit more ductility. The latter behavior is not exhibited by typical FRP laminates; therefore, for analytical purposes, the SRP textiles behavior can be approximated by a bi-linear strain-hardening stress-strain relationship as shown in Figure 4 and expressed by Equation (4a,b). Such a relationship is adopted in the proposed analysis.
(4a)$$\sigma_{f}=E_{f,1}\varepsilon \;\mathrm{for}\;\varepsilon <\varepsilon_{fy}$$
(4b)$$\sigma_{f}=\sigma_{fy}+E_{f,2}\left(\varepsilon -\varepsilon_{fy}\right)\;\mathrm{for}\;\varepsilon_{fy}<\varepsilon <\varepsilon_{fu}$$

The adhesive layer bonding the SRP to the substrate is mainly subjected to planar shear stress, which results in Mode II fracture. Fracture in Mode II is commonly represented by a bi-linear law composed of an ascending branch until the maximum shear stress is reached, followed by a descending or softening branch until complete loss of strength [32,33,34]. After the ultimate deformation or slip, $\lambda_{u}$, is reached or exceeded, full delamination is assumed to have occurred. Mathematically, the bond-slip relationship can be expressed as
(5a)$$\tau =k_{s}\lambda \;\mathrm{for}\;\lambda <\lambda_{0}$$
(5b)$$\tau =k_{b}\left(\lambda_{u}-\lambda \right)\;\mathrm{for}\;\lambda_{0}<\lambda <\lambda_{u}$$
(5c)$$\tau =0\;\mathrm{for}\;\lambda >\lambda_{u}$$

With reference to Figure 5, in Equation (5a–c), $\lambda_{0}$ represents the slip corresponding to the maximum shear resistance $\tau_{\max}$ while $k_{s}$ and $k_{b}$ are the shear stiffnesses of the interface corresponding, respectively, to the slope of the ascending and descending branches in the interfacial shear-slip relationship. 

Finally, in conformity with previous studies [14], concrete in tension is assumed to be a brittle linear elastic material whose stress-strain relationship can be written as
(6)$$\sigma_{c}=E_{c}\frac{du_{c}}{dx}$$
where $E_{c}$ represents the concrete elastic modulus. Although concrete is known to exhibit tension stiffening, this characteristic is not germane to the current analysis because failure in the cases with which the present analysis is concerned is almost always initiated by interfacial debonding. 

## 3. Proposed Analytical Model

### 3.1. Monotonic Load Case

In this section, the complete debonding process in a single lap shear test subjected to a monotonically increased load $P_{i}$ will be analyzed using a stage-by-stage approach. In particular, the proposed model permits the investigation of the axial strain and shear stress distribution, the interfacial slip along the bonded length, and the load-deformation curve of the bonded portion of SRP.

#### 3.1.1. Elastic or Ascending Stage

For low load levels, the shear stress along the bonded length will be in the elastic or ascending stage. In this case, based on the constitutive relationship of the SRP textile Equation (7) and its substitution in Equations (1) and (4a), one can write
(7)$$\frac{d\sigma_{f}}{dx}=E_{f,1}\frac{d^{2}u_{f}}{dx^{2}}$$
(8)$$E_{f,1}t_{f}\frac{d^{3}u_{f}}{dx^{3}}-k_{S}\left(\frac{du_{f}}{dx}-\frac{du_{c}}{dx}\right)=0$$

Based on Equation (3), the axial stress at the concrete substrate level can be written as
(9)$$\sigma_{c}=-\sigma_{f}\frac{t_{f}b_{f}}{t_{c}b_{c}}$$

Considering the linear elastic behaviour of concrete per Equation (6), the strain in the concrete can be expressed as
(10)$$\frac{du_{c}}{dx}=-\frac{\sigma_{f}A_{f}}{E_{c}A_{c}}$$

Finally, substituting Equation (10) into Equation (8) results in the following second-order homogeneous differential equation
(11)$$\varepsilon_{f}^{\prime \prime }-\omega_{I,1}^{2}\varepsilon_{f}=0\;\mathrm{with}\;\omega_{1,I}^{2}=k_{s}\left(\frac{1}{E_{f,1}t_{f}}+\frac{b_{f}}{E_{c}A_{c}}\right)$$
where the relation $\varepsilon_{f}=\frac{du_{f}}{dz}$ represents the axial strain of the SRP textile.

The general solution, $\varepsilon_{f}(x)$, of the above differential equation is
(12)$$\varepsilon_{f}{}_{I,1}(x)=A_{I,1}e^{\omega_{I,1}x}+B_{I,1}e^{-\omega_{I,1}x}$$

In which the two unknown constants can be determined by insertion into Equation (12) the relevant boundary conditions, expressed as Equation (13a,b), that is
(13a)$$N_{f}\left(L_{f}\right)=P$$
(13b)$$\varepsilon_{f}{}_{I,1}(0)=0$$

#### 3.1.2. Softening or Descending Stage

After the interfacial slip exceeds $\lambda_{0}$, the softening stage commences along the bonded length, starting from the loaded end and propagating towards the unloaded end. To model this stage, Equation (4b) is inserted in Equation (1), resulting in
(14)$$E_{f,1}t_{f}\frac{d^{3}u_{f}}{dx^{3}}-k_{b}\left(\frac{du_{c}}{dx}-\frac{du_{f}}{dx}\right)=0$$

Inserting Equation (10) in Equation (14) leads to the following governing differential equation in the latter case
(15)$$\varepsilon_{f}^{\prime \prime }+\omega_{1,II}^{2}\varepsilon_{f}=0\;\mathrm{with}\;\omega_{II,1}^{2}=k_{b}\left(\frac{1}{E_{f,1}t_{f}}+\frac{b_{f}}{E_{c}A_{c}}\right)$$

The solution of Equation (15) can be written as
(16)$$\varepsilon_{f_{\;II,1}}\left(x\right)=A_{II,1}\cos\left(\omega_{II,1}x\right)+B_{II,1}sin\left(\omega_{II,1}x\right)$$

In this stage, the bonded length must be divided into two regions. The first is the region from the unloaded end of the SRP to the point where *λ* ≤ *λ*_0_, while the second region comprises the remaining length. The bond-slip characteristics of the two regions are governed by the ascending and descending shear stress-slip relationship in Figure 4. 

The general solution of the governing equation is given by Equation (12) for the elastic region and by Equation (16) for the softening region. The solutions of these two equations produce four constants of integration, which can be solved by enforcing the four boundary conditions given by Equation (17a–d):(17a)$$N_{f}\left(L_{f}\right)=P$$
(17b)$$\varepsilon_{f}{}_{I,1}(x_{a})=\varepsilon_{f}{}_{II,1}(x_{a})\;(\mathrm{due}\;\mathrm{to}\;\mathrm{the}\;\mathrm{continuity})$$
(17c)$$\varepsilon_{f\;I,1}^{\prime }(x_{a})=\varepsilon_{f\;II,1}^{\prime }(x_{a})\;(\mathrm{due}\;\mathrm{to}\;\mathrm{the}\;\mathrm{continuity})$$
(17d)$$\varepsilon_{f}{}_{I,1}(0)=0$$

Note, the parameter distinguishing region one from two is $x_{a}$, which can be determined by enforcing either of the following conditions:(18a)$$\lambda (x_{a})=\lambda_{0}$$
(18b)$$\tau (x_{a})=\tau_{\max}$$

#### 3.1.3. Debonding with the SRP Remaining Elastic

Besides considering the interfacial conditions leading to the above stage, two scenarios must be considered depending on the stress level in the SRP textile, i.e., whether it is in the elastic or plastic state before the initiation of debonding. In the following analysis, it is first assumed that the SRP stress is below its yield stress and is consequently behaving elastically. 

In this scenario, the applied load can be increased until ultimate slip $s_{u}$ is reached, and the debonding process begins at the loaded end of the SRP textile. The pertinent debonding stage can be represented by the following equation:(19)$$\varepsilon_{f\;deb}^{\prime }(x)=0$$

The solution of Equation (19) is a constant as given in Equation (20):(20)$$\varepsilon_{f\;deb}(x)=A_{deb}$$

Integrating Equation (20) one can find the axial displacement of the textile in the debonding zone as
(21)$$u_{f}{}_{deb}(x)=A_{deb}x+B_{deb}$$

This stage is characterized by the presence of three regions along the bonded length: (a) the region governed by elastic shear stress-slip response, (b) the region governed by the softening branch of the shear stress-slip relationship, and (c) the SRP textile being completely separated from the concrete substrate.

The relevant solutions for the above three cases are given by Equations (12), (16) and (20). Due to the presence of three zones, there will be five integration constants that need to be determined using the relevant boundary conditions as follows: (22a)$$N_{f}\left(L_{f}\right)=P$$
(22b)$$\varepsilon_{f}{}_{II,1}(x_{p})=\varepsilon_{f}{}_{deb}(x_{p})$$
(22c)$$\varepsilon_{f}{}_{I,1}(x_{a})=\varepsilon_{f}{}_{II,1}(x_{a})\;(\mathrm{due}\;\mathrm{to}\;\mathrm{the}\;\mathrm{continuity})$$
(22d)$$\varepsilon_{f\;I,1}^{\prime }(x_{a})=\varepsilon_{f\;II,1}^{\prime }(x_{a})\;(\mathrm{due}\;\mathrm{to}\;\mathrm{the}\;\mathrm{continuity})$$
(22e)$$\varepsilon_{f\;I,1}(0)=0$$

It can be observed in Equation (22a–e) that the three regions can be identified by determining the two position parameters $x_{a}$ and $x_{p}$, where $x_{a}$ represents the point along the interface where the interfacial shear stress equals the maximum shear resistance, *τ*_max_, while $x_{p}$ represents the point where resistance is exhausted and drops to zero, thus initiating the FRP/SRP separation.

The four additional conditions that need to be satisfied to enforce the above scenario are:(23a)$$\lambda (x_{a})=\lambda_{0}$$
(23b)$$\tau (x_{a})=\tau_{\max}$$
(23c)$$\lambda (x_{p})=\lambda_{u}$$
(23d)$$\tau (x_{p})=0$$

#### 3.1.4. Debonding after the SRP Entering the Plastic State

In the previous section, the equations governing SRP textile debonding under the scenario of the textile remaining elastic until complete debonding were developed. In the following, the alternative scenario is considered where the SRP enters the plastic state before the incidence of total interfacial separation. In this case, in view of Equation (4), one can write
(24)$$\frac{d\sigma_{f}}{dx}=E_{f,2}\frac{d^{2}u_{f}}{dx^{2}}$$

Substituting Equation (24) in Equation (1) and assuming the softening part of the shear stress-slip relationship (Equation (4b)), one can write
(25)$$E_{f,2}t_{f}\frac{d^{3}u_{f}}{dx^{3}}-k_{b}\left(\frac{du_{c}}{dx}-\frac{du_{f}}{dx}\right)=0$$

Equation (25) can be recast as follows, which is the governing equation of the scenario under consideration.
(26)$$\varepsilon_{f}^{\prime \prime }+\omega_{II,2}^{2}\varepsilon_{f}+\eta_{II,2}=0$$

The solution of Equation (26) is given as
(27)$$\varepsilon_{f_{\;II,2}}\left(x\right)=A_{II,2}\cos\left(\omega_{II,2}x\right)+B_{II,2}sin\left(\omega_{II,2}x\right)-\frac{\eta_{II,2}}{\omega_{II,2}{}^{2}}$$

With
(28a)$$\omega_{II,2}^{2}=k_{b}\left(\frac{1}{E_{f,2}t_{f}}+\frac{b_{f}}{E_{c}A_{c}}\right)$$
and
(28b)$$\eta_{II,2}=k_{b}\left(\frac{b_{f}\sigma_{f_{y}}}{E_{c}A_{c}E_{f,2}}-\frac{b_{f}\varepsilon_{f_{y}}}{E_{c}A_{c}}\right)$$

The preceding set of equations can be used to obtain the stress and deformation of the concrete substrate, the SRP textile and the SRP-concrete interface for various failure modes that are observed in single-lap shear tests. 

#### 3.1.5. Effective Bond Length

The effective bond length represents a key parameter in the study of the delamination of reinforcing laminates from their substrate. It is well known that the effective bond length may be shorter than the provided bond length. Hence, knowledge of the former length is necessary for determining the debonding load or ultimate capacity. 

In the proposed model, the effective bond length, *L_eff_*, is given by
(29)$$L_{eff}=L_{eff,I}+L_{eff,II}=\frac{1}{\omega_{I,1}}+\frac{\pi }{2\omega_{II,1}}$$

As can be observed in Equation (29), the effective bond length is composed of the sum of the length of the elastic and softening regions, denoted as $L_{eff,I}$ and $L_{eff,II}$, respectively. Furthermore, it is important to underline that the effective bond length is a function of several parameters, including the thickness and stiffness of the reinforcing textile, the concrete prism mechanical properties, and the stiffness of the elastic and softening branches of the interface.

### 3.2. Cyclic Load Case

In practice, a retrofitted concrete element may be subjected to cyclic loads as in the case of bridges under traffic load or in the case of structures subjected to earthquake, wind or other fluctuating dynamic loads. The response of the retrofitted element, as idealized in the single-lap shear test in which the laminate is subjected to cyclic tensile load, as in Figure 6, is modelled analytically here and the relevant governing equations and their solutions are presented below. 

The cohesive stress-separation law is defined by Equation (5) as presented earlier.

The total fracture energy is given by
(30)$$G_{II}=\int_{0}^{\lambda_{u}}\tau dx$$
where $G_{II}$ represents the total energy and is equal to the sum of the elastic and inelastic energies represented by area under the shear stress-slip curve, and denoted by $G_{E,II}$ and $G_{S,II}$, respectively, as in Equation (31): (31)$$G_{II}=G_{E,II}+G_{S,II}$$

García-Collado et al. [35] named $G_{E,II}$ the forward region and $G_{S,II}$ the wake zone on the basis of the concepts of Ritchie [36], who essentially defined two possible classes of fatigue mechanisms: intrinsic and extrinsic. 

Under the unloading cycle, two possible scenarios can be envisaged along the bonded length: (a) the interfacial shear stress being less than the maximum shear resistance, (b) the slip exceeding the slip corresponding to the maximum shear resistance, $\lambda_{0}$. The first case involves only elastic energy, less than or equal to $G_{E,II}$, which is fully released/recovered after complete unloading and no damage or permanent deformation occurs under this scenario. The second scenario, on the other hand, involves both $G_{E,II}$ and $G_{S,II}$ where the former energy is completely recoverable as under scenario one, while the latter is fully dissipated. In the proposed model, in scenario two, loading and unloading are assumed to occur along a line parallel to the ascending part of the shear stress-slip curve as shown in Figure 7. This means that unloading from a certain stress level to zero and reloading the specimen back to the same stress level does not cause any additional damage beyond that incurred at the start of the unloading process. In other words, hysteresis is assumed to be negligible. 

The preceding unloading/reloading process can be expressed as follows
(32a)$$k_{C}(x)=k_{S}\;\mathrm{for}\;\lambda \leq \lambda_{0}$$
(32b)$$k_{C}(x)=k_{S}{\left[\frac{\lambda_{u}-\lambda \left(x\right)}{\lambda_{u}-\lambda_{e}}\right]}^{2}\;\mathrm{for}\;\lambda >\lambda_{e}$$

The expression Equation (32) can be obtained after some manipulations considering Equation (5a,b), the elastic energy $G_{E,II}$, and if the elastic area does not change under any unloading/reloading process. 

The shear stress-slip law can be evaluated in the following way:(33)$$\tau =k_{C}\lambda$$

It can be observed in Figure 7 that upon full unloading, permanent damage occurs, which is represented by the permanent slip. In the post-elastic state, the slip at any point located at distance x from the laminate unloaded end can be determined using
(34a)$$\lambda \left(x\right)=Ae^{\left(w_{C}x\right)}+Be^{\left(-w_{C}x\right)}$$
(34b)$$\lambda \left(x\right)=A+Bx+Cx^{2}$$
where A, B and C are constants of integration and the coefficient *w*_x_ is given by
(35)$$w_{C}^{2}=k_{C}b_{f}\left(\frac{A_{f}E_{f}+A_{c}E_{c}}{A_{f}E_{f}A_{c}E_{c}}\right)$$

Knowing the interface deformation function, $\lambda \left(x\right)$, the total strain can be obtained as
(36)$$\varepsilon \left(x\right)=\frac{d\lambda }{dx}$$

The total strains can be expressed as sum of its elastic, $\varepsilon_{e}$, and plastic $\varepsilon_{p}$ parts as
(37)$$\varepsilon_{t}\left(x\right)=\varepsilon_{e}\left(x\right)+\varepsilon_{p}\left(x\right)$$

The tensile force resisted by the laminate–concrete ensemble can be evaluated by integrating the shear stresses acting on the interface as follows
(38)$$F=b_{f}\Delta x\sum \tau \left(x\right)$$
with Δx being the distance between two points along the interface with average shear stress *τ*(*x*). 

## 4. FEM Model

In this study, a finite element model was made and analyzed to simulate the aforementioned single-lap shear test. Due to symmetry, shown in Figure 8, only half of the specimen was discretized by an assemblage of 3D finite elements as shown in Figure 9. Using the software Abaqus [37], a fully implicit integration scheme was adopted in the analysis 

Eight-node quadratic brick elements (C3D8), each having side lengths of 5 mm, were used to discretize the concrete block, while four-node 2D shell elements (S4) with side lengths of 5 mm were used to discretize the SRP strip. Further details of the adopted mesh are provided in Table 1. The constitutive models adopted for the concrete elements and FRP sheet were isotropic and their mechanical characterizing parameters are shown in Table 2a. 

To simulate the adhesive layer, a cohesive law, representing the damage between the laminate and concrete, was adopted, which is characterized by linear ascending and descending parts as illustrated in Figure 5 in relation to the law used to represent Mode II fracture. The values of the relevant parameters of this law, as used in the current analysis, are summarized in Table 2b.

For checking convergence of the solution, the force control criterion was selected while the analysis was performed via displacement control. The Newton–Raphson method was adopted to solve the non-linear system of equations, with nonlinearity being caused by yielding of the SRP laminate and/or by the phenomena associated with the fracture process. 

In Figure 10, the results of the finite element analysis and analytical model are compared in terms of the load-slip relationship. The laminate–concrete system was also subjected to cyclic load where unloading was allowed when the slip first reached 0.1 mm and then 0.2 mm. 

One can observe in Figure 10 that under monotonic load, the FEM and the proposed model results are practically identical, but the unloading responses under cyclic load are different. The difference is due to different damage definitions used in the FEM and the proposed analytical model. For clarity, the loading–unloading law used in FEM analysis is shown in Figure 11. In the latter law, damage is defined in terms of the rate of stiffness degradation after damage initiation, triggered by the maximum interfacial shear stress exceeding the interface maximum shear resistance. A scalar damage index, D, is applied, which is assumed to represent the overall damage in the material and captures the combined effects of all the active damage mechanisms. It initially has a value of zero, but as damage is accumulated, *D* is assumed to monotonically evolve from 0 to 1. The post-damage stress component, *τ_s_*, is calculated as
(39)$$\tau_{s}=\left(1-D\right){\bar{\tau }}_{s}$$
where ${\bar{\tau }}_{s}$ is the shear stress component predicted by the elastic stress-slip behavior without damage. This law may correctly predict the post-peak stress and the concomitant damage level, but the associated deformation is not correctly captured while the proposed model is expected to accurately predict all three quantities. It should be noted that if the evolution of interfacial damage is associated with stress or load redistribution, application of Equation (12) may not predict the correct failure load. 

It can be noticed in Figure 11 that the interface constitutive law adopted in the FEM analysis does not comprise any permanent slip upon unloading, which does not comport with empirical evidence while the proposed model, as shown in Figure 7, furnishes the extent of permanent slip upon unloading.

## 5. Comparison with Experimental Results

### 5.1. Monotonic Loading Tests

To ascertain the validity of the assumptions in the proposed model as well as the model’s accuracy and robustness, several analyses are made, and the results are compared with the companion experimental data obtained from the literature. The cases investigated involve (a) CFRP plate-concrete specimen with CFRP having a thickness of 1.016 mm [38]; (b) SRP textile-concrete specimens, with SRP having a thickness of 0.084 mm, 0.254 mm or 0.381 mm, and designated as low-density (LD), medium-density (MD) and high-density (HD) SRP, respectively [1]; (c) MD textile bonded to concrete block (CB) or MD textile bonded to tuff (TU) block specimens [39]. In all three cases, in the physical experiments, single-lap shear tests were performed under monotonically increasing load until failure. 

Relevant properties to the concrete–laminate system are given in Table 3. As the values of certain parameters in the experiments were not reported here, for the purposes of performing the analysis, they are assumed based on suggested values in the literature or using engineering judgment. The assumed values are shown in italics in Table 3. 

Figure 12 shows the experimental data and the presently computed load-slip curves for the CFRP plate bonded to the concrete substrate in Case (a). On the other hand, for the same case, Figure 13 shows for the CFRP laminate the experimental and presently computed strain distribution along its bonded length.

In Figure 14, the results for Case (b) are shown in terms of applied load versus slip. It is important to remark that in Case (b) the parameters of the bond-slip law were calibrated by Ascione et al. [40] using the experimental results in [1]. 

In Figure 15, analytical results for Case (c) are compared with the companion experimental data. It can be observed that, in all cases the analytical results are in good agreement with the corresponding experimental data. The observed differences are primarily due to the natural variability in the properties of the substrate and the quality of the workmanship when bonding the laminate to the substrate. In practical applications, such variabilities can be accounted for by reliability-based resistance factors. 

Finally, the proposed Equation (29) is used to compute the effective bonded length of the specimen’s strength with LD, MD, and HD SRP textiles, and the computed values are compared with the corresponding experimental values reported by Ascione et al. [1]. They reported the effective length for the specimens with LD textile in the range of 60–90 mm, for those with MD textiles in the range of 120–150 mm and for HD textiles in the range of 150–200 mm. Here, the textiles’ mechanical properties given in [1] and the parameters of the interface bond-slip law given in [40] are used to compute effective bond lengths of 89 mm, 154 mm and 188 mm for the specimens made with LD, MD and HD textile, respectively. It can be observed that the computed values are practically all within the relevant observed ranges.

For comparison, Italian guideline CNR-DT 200 R1/2013 [41] provides a semi-empirical expression for calculating the required bonded length of FRP laminates attached to concrete substrate by a suitable adhesive. Based on that expression, for the above test specimens, the computed values are 77 mm, 133 mm and 163 mm for the specimens made with LD, MD and HD textile, respectively. Based on these values, the expression in the Italian guidelines also gives reasonable values for the effective bonded length of SRP textile-concrete assemblages. 

### 5.2. Cyclic Load Case

To validate the proposed analytical model and to evaluate its performance in the case of test specimens subjected to cyclic loading, in the following, some physical tests reported in the literature are analyzed and the analytical results are compared with the corresponding experimental data. 

#### Nigro et al. [27] Tests

These investigators reported the results of monotonic and cyclic load tests on concrete prisms strengthened with CFRP sheets. In particular, the prisms were 150 mm wide, 200 mm high and 500 mm long. The concrete cylinder mean strength, $f_{cm}$, was 22.5 MPa. The CFRP strip was 100 mm wide and it had a bonded length of 400 mm, with an elastic modulus and strength of 216 GPa and 3240 MPa, respectively. The authors tested duplicate specimens to gauge the consistency of the interface performance.

In Figure 16, the experimental load-slip curves for three repeat specimens, designated as SM_1, SM_2 and SM_3, subject to monotonic load, are plotted, together with the companion computed curve obtained by using the proposed model. In the analytical model, based on the results in [27], the maximum slip, corresponding to total debonding, was taken as 0.2 mm for monotonic loading and 0.21 for cyclic loading. 

It can be observed that the computed results are in good agreement with the corresponding experimental data and the differences among them can be mainly ascribed to the random variations in material properties and workmanship in specimen preparation. 

The latter investigators tested two nominally identical specimens, designated as SC_4 and SC_5, under cyclic load. These specimens were analyzed using the proposed model and the results are shown in Figure 17 and Figure 18, respectively.

It can be observed that the results of the proposed analytical model agree well with the corresponding experimental results throughout the loading, unloading and reloading process. Consequently, the model seems sufficiently robust and can provide a reasonable estimate of the debonding load for specimens in single-lap shear tests subjected to monotonic or cyclic load. 

## 6. Parametric Analysis Using the Proposed Model

### 6.1. Static Load Case

Using the proposed model, in this section, debonding of the steel textile from the concrete substrate is further analyzed by investigating the effect of some key parameters on the debonding load and deformations. To avoid premature debonding due to insufficient bonded length, a bonded length equal to 300 mm is chosen, which is guided by experimental data in 0. The concrete prism dimensions are held fixed at 200 × 150 × 400 mm (width × height × length) for all the analyzed specimens. The thickness of steel textile is set equal to 0.381 mm, its width to 100 mm and the maximum interfacial shear resistance, *τ*_max_, equal to 3.5 MPa. 

Figure 19, Figure 20, Figure 21 and Figure 22 show the textile axial strain, its tensile stresses along the bonded length, and the interfacial shear stress and slip evolution, respectively, under increasing applied load. In Figure 19, it can be noticed that the curvature of the axial strain curve changes from concave to convex as the interfacial shear stress begins to exceed the associated shear resistance and the shear resistance enters the descending or softening branch of the shear-slip constitutive law. As the load is further increased beyond that eliciting the maximum shear stress, the strain over a large part of the bonded length becomes constant, which signifies debonding over that part. It can be also noticed that the debonding initiates at approximately 90% of the maximum or failure load; hence, the debonding process is relatively sudden. It is also important to mention that up to 90% of the failure load, the effective bonded length is about 150 mm, which is only half of the provided bonded length. 

Figure 20 shows the evolution of axial stress in the textile along its bonded length under increasing load. As expected, the stress follows a practically identical pattern as the associated strain due to the textile stress being less than the SRP yield stress. Since in this analysis the textile was assumed to be HD, such high-strength textile will not generally yield because failure will be initiated by loss of interfacial shear resistance and debonding before the initiation of yielding.

Figure 21 shows the variation of slip along the bonded length under various load levels. This figure again demonstrates that up to 90% of the failure load, the slip is relatively small, and it slowly increases as the load is increased, but as the load is increased beyond the above level, the slip sharply increases and soon after, failure occurs. Consequently, in practical applications, the proposed model can be used to determine the design service and ultimate load for a certain combination of FRP/SRP, adhesive and substrate while avoiding sudden or unexpected failure. 

Figure 22 shows the distribution of interface shear stress along the bonded length under increasing applied load. Notice how the location where maximum shear stress occurs begins to shift along the bonded length. In addition, it can be observed that maximum shear stress is first reached at the laminate loaded end at only 40% of the failure load of the assemblage; hence, the FRP/SRP assemblage has substantial load redistribution capacity. For design purposes, shear stress distribution graphs, similar to the ones in Figure 22 can be plotted for various SRP/substrate combinations to obtain the average interfacial shear at failure, a quantity that is normally obtained from physical tests and used in design. 

Figure 23 shows the variation of the normalized axial stress in the steel textiles of different densities along the bonded length. Note that the stress is expressed as fraction of the ultimate strength of the relevant textile. Since the textile density is a function of its thickness and since its ultimate load capacity is also a function of its thickness, the higher the textile density, the higher its ultimate load capacity. For this reason, the LD textile achieved the greatest fraction of its ultimate strength compared to the MD and HD textiles. Despite this higher stress in LD, the highest axial force was resisted by HD, MD and LD, respectively. Finally, it would appear that the debonded interfacial length is practically independent of the textile density.

### 6.2. Cyclic Load Case 

In this section, some analyses, using the proposed model, are performed to gain greater insight into the behavior and strength of FRP/SRP retrofitted concrete elements subjected to cyclic load. It must be emphasized that the analyzed cases are supposed to represent physical single-lap shear tests. The maximum cyclic load to be applied was established by considering the maximum or failure load reached in analogous specimens under monotonic loading. Specifically, the load cycles involved the following:

1° Cycle: increase the load from zero to 50% of the maximum load, then begin unloading to zero

2° Cycle: reload from zero to 70% of the maximum load, then begin unloading to zero

3° Cycle: reload from zero to 85% of the maximum load, then begin unloading to zero

The analytical results in terms of force and displacement are depicted in Figure 24.

In Figure 25 and Figure 26 the permanent slip and the plastic strain distributions are plotted along the bonded length at the end of each of the above-described three load cycles. Notice that with each cycle, as the load at which unloading occurs is increased, the permanent slip and plastic strain in the textile increase accordingly, but the increase is not linearly proportional. The last two figures also show that as unloading is carried out from higher load levels, the permanent slip and plastic strain propagate to a larger portion of the bonded length. Although these responses are expected, the current model provides a quantitative measure of the extent of plastic deformations in the RP/SRP laminate and its interface with the concrete substrate. In practice, to avoid unexpected failure, the computed values by the current model can be compared to the recommended limits for these quantities specified in design guidelines and standards. 

## 7. Conclusions

In this paper, a novel and relatively simple analytical model is proposed to analyze the interfacial behavior and strength of FRP/SRP laminates or textiles bonded by an adhesive to concrete or other similar substrate. The model is applicable to elastic or elasto-plastic strain-hardening laminates subjected to monotonic or cyclic loading. The model reliability and accuracy are verified by comparing its results with the corresponding experimental data and with the results of three-dimensional finite element analysis. These comparisons are made by analyzing concrete prisms retrofitted with FRP/SRP laminates subjected to monotonic or cyclic loading in single-lap shear tests. Based on the comparisons and subsequent parametric studies, the following principal conclusions are reached:

The proposed analytical model and its associated bond-slip law can predict the interfacial response of FRP/SRP laminates bonded to concrete, or other similar, substrates subjected to monotonic or cyclic load.

The model can accurately predict the ultimate load and the associated slip of FRP/SRP retrofitted prisms tested in single-lap shear.

The model can trace the evolution of the debonding zone along the bonded length of laminate and can provide the corresponding interfacial shear stress distribution and the laminate axial stress and strain along the entire bonded length at any load level up to failure.

The model can be used to assess, under a given monotonic or cyclic load, the extent of interfacial damage incurred by the retrofitted element. In assessing the damage, a damage index based on permanent interfacial slip can be applied. 

## Figures and Tables

**Figure 1 materials-15-08690-f001:**
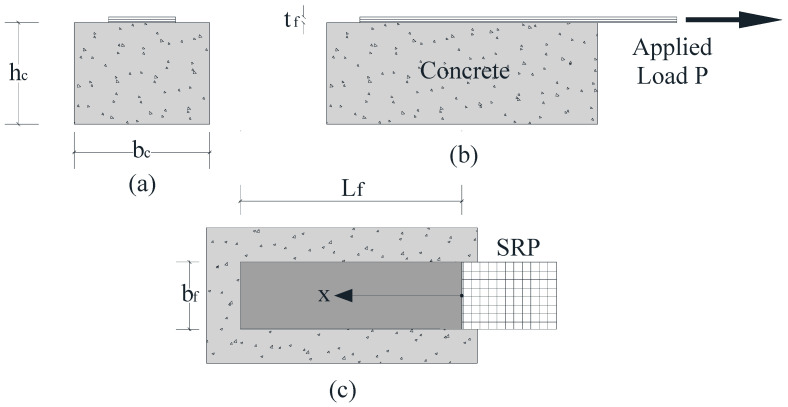
Single lap shear test specimen geometry: (**a**) front view; (**b**) side view; (**c**) top view.

**Figure 2 materials-15-08690-f002:**
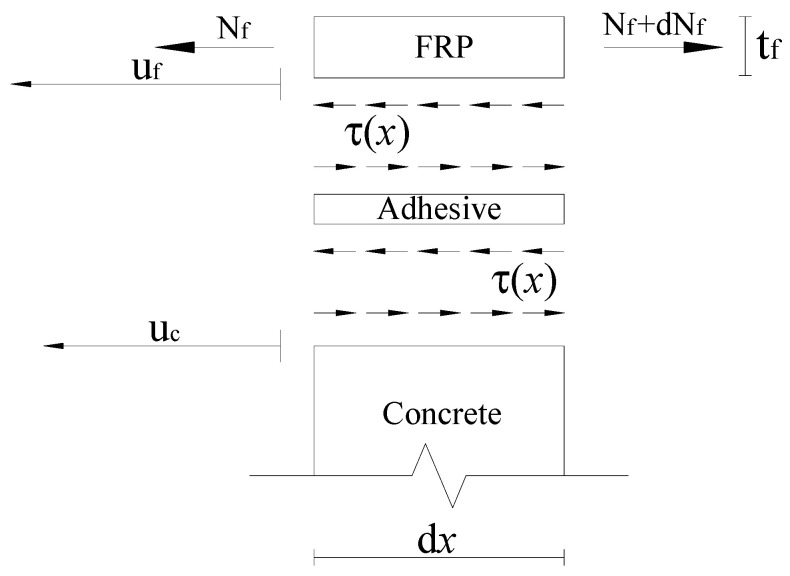
Equilibrium of an infinitesimal element of length *dx* along the specimen.

**Figure 3 materials-15-08690-f003:**
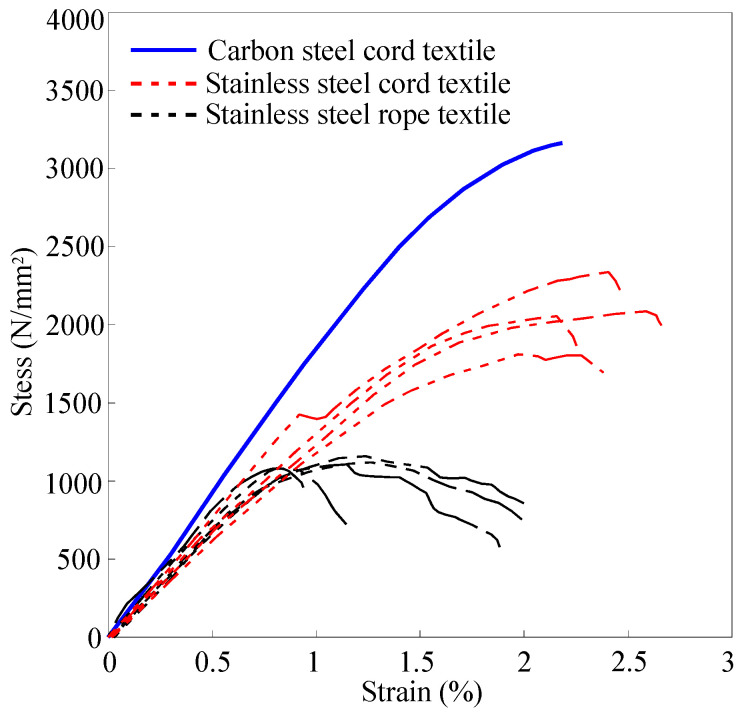
Tensile stress-strain relationship different SRP textiles.

**Figure 4 materials-15-08690-f004:**
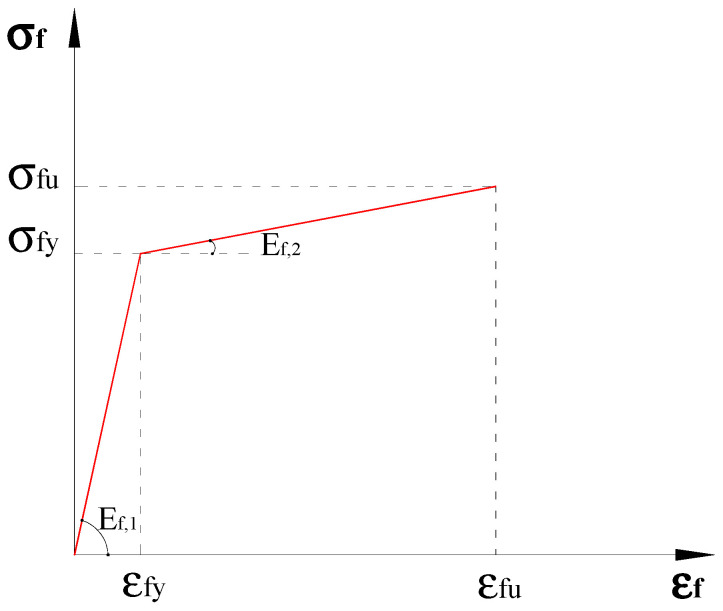
Steel textile stress-strain relationship.

**Figure 5 materials-15-08690-f005:**
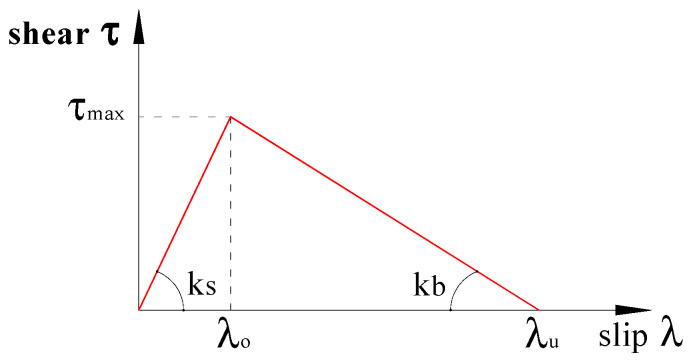
Shear stress-slip relationship of the adhesive layer at the interface.

**Figure 6 materials-15-08690-f006:**
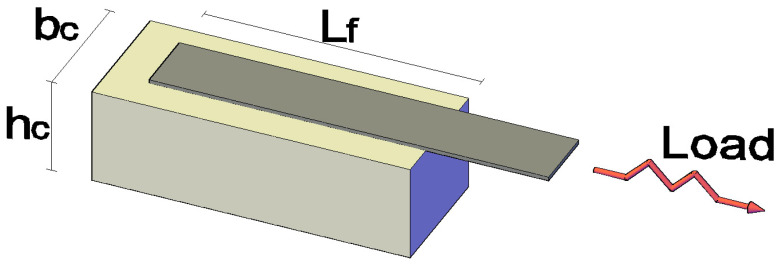
Single-lap shear test geometry under cyclic load.

**Figure 7 materials-15-08690-f007:**
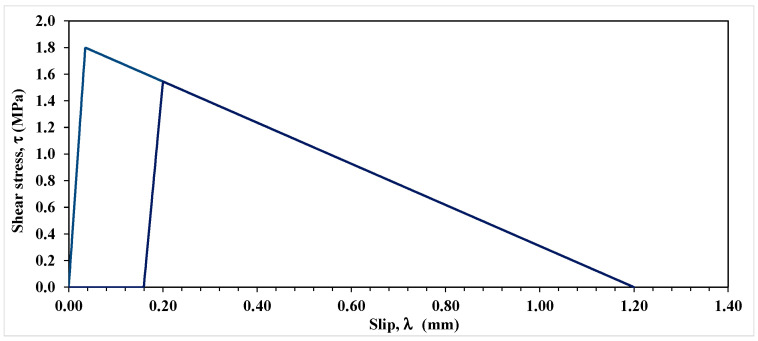
Representation of the cohesive shear stress-slip law for the loading-unloading-reloading process.

**Figure 8 materials-15-08690-f008:**
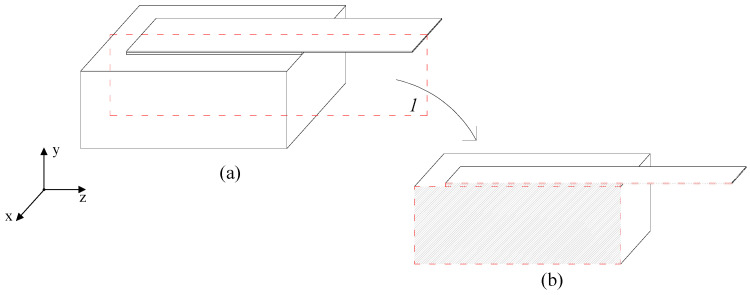
Symmetry plane: (**a**) single-lap shear specimen, (**b**) symmetrical part.

**Figure 9 materials-15-08690-f009:**
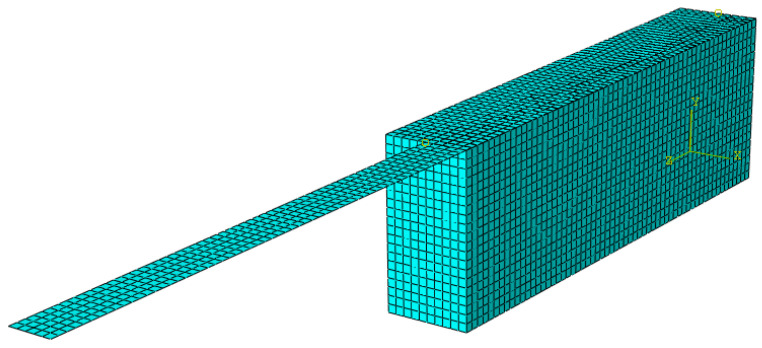
3D FEM model of single-lap shear test.

**Figure 10 materials-15-08690-f010:**
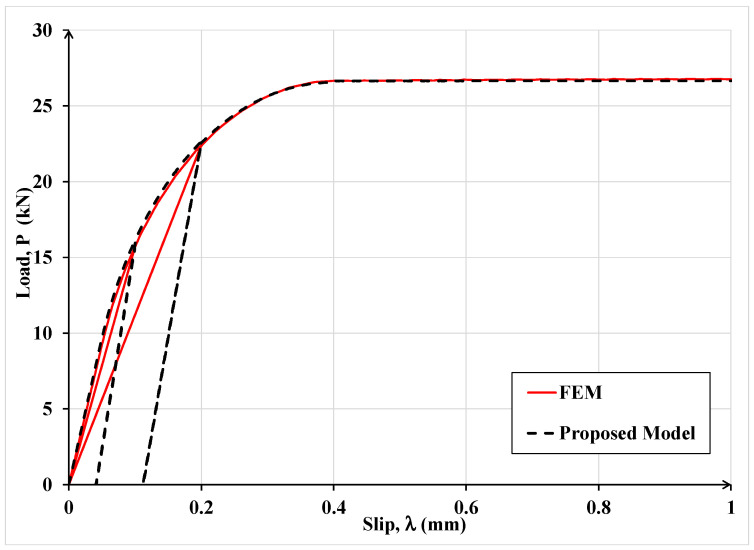
Comparison of the FEM and the proposed analytical model results for simulating the single-lap shear test under cyclic load.

**Figure 11 materials-15-08690-f011:**
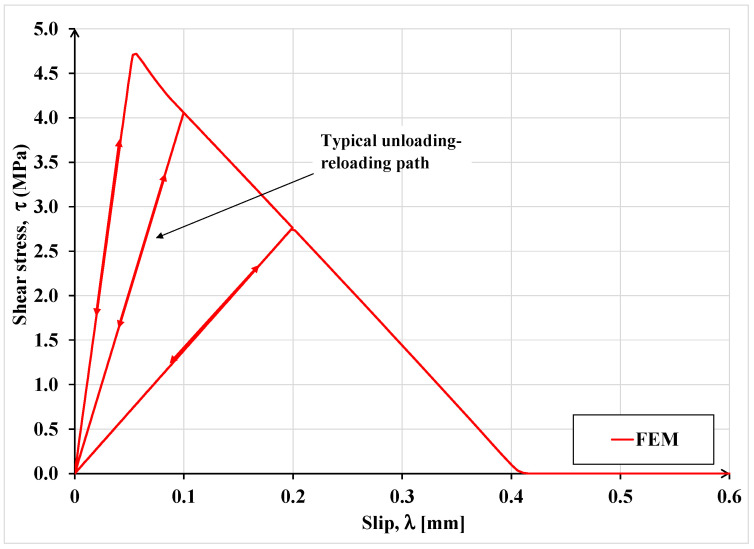
Cohesive stress-separation law applied in the FEM model for representing loading-unloading-reloading sequences.

**Figure 12 materials-15-08690-f012:**
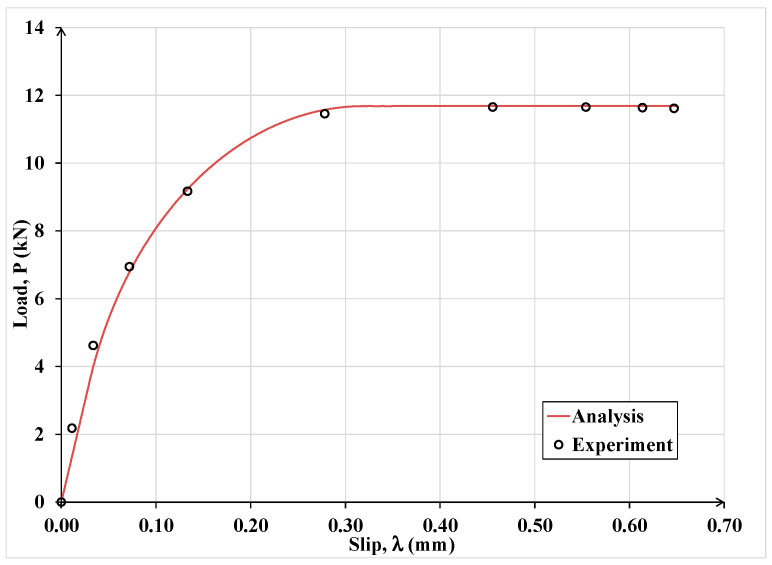
Current analytical versus experimental load-slip curve of the CFRP for Case (a).

**Figure 13 materials-15-08690-f013:**
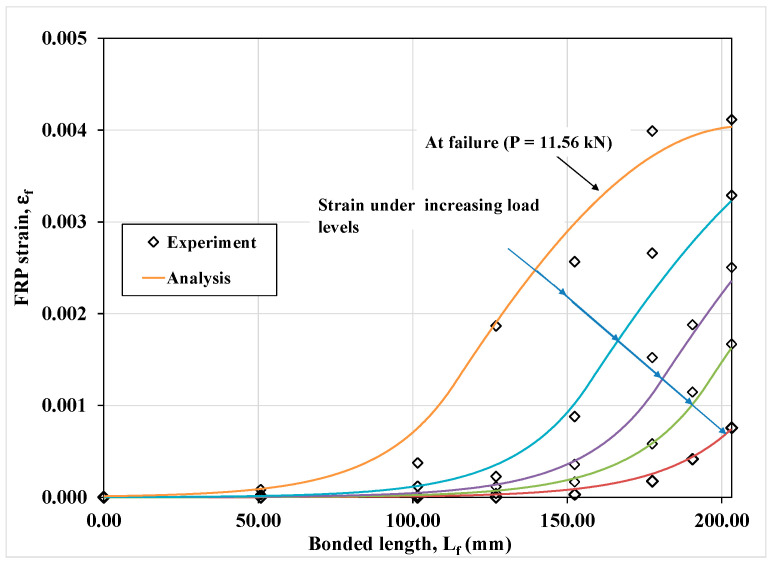
Analytical versus experimental CFRP strain distribution along the bonded length in Case (a).

**Figure 14 materials-15-08690-f014:**
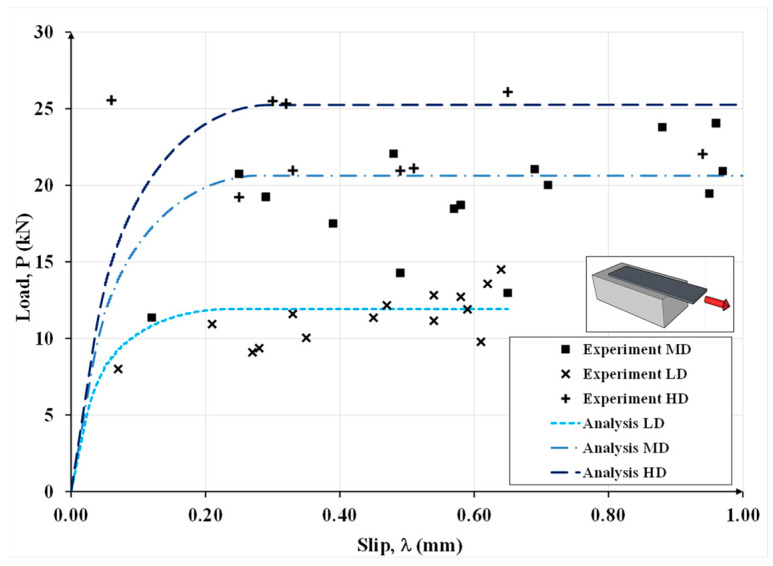
Experimental and analytical load-slip curves for the test specimens with low-density, medium-density and high-density steel textiles in Case (b).

**Figure 15 materials-15-08690-f015:**
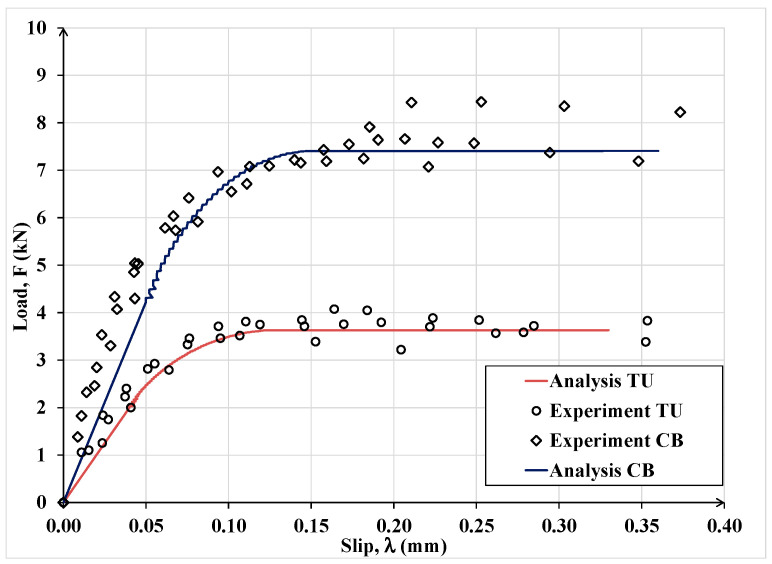
Experimental versus analytical load-slip curves for the test specimens in Case (c) involving MD steel textile bonded to concrete and tuff substrates.

**Figure 16 materials-15-08690-f016:**
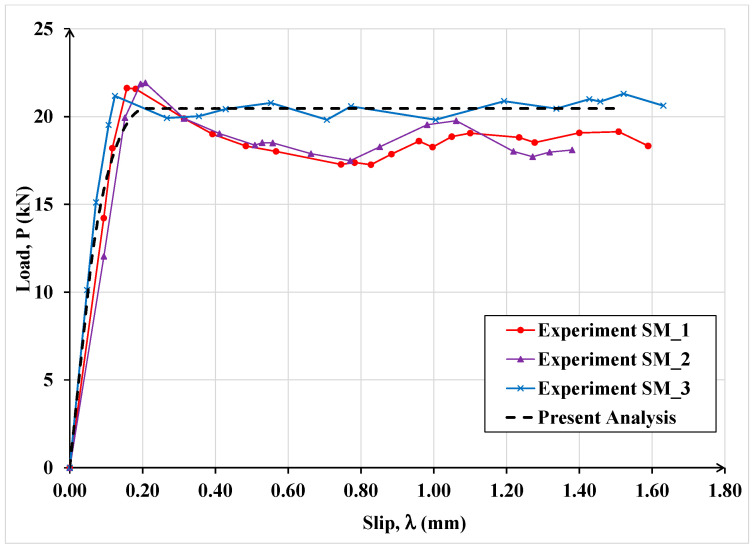
Comparison between experimental and analytical results under monotonic load.

**Figure 17 materials-15-08690-f017:**
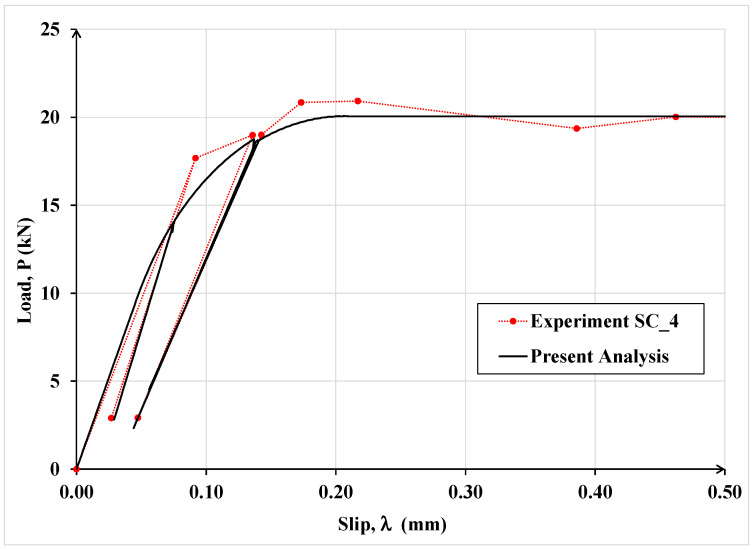
Comparison of experimental and analytical load-slip curves for specimen SC_4 subjected to cyclic load.

**Figure 18 materials-15-08690-f018:**
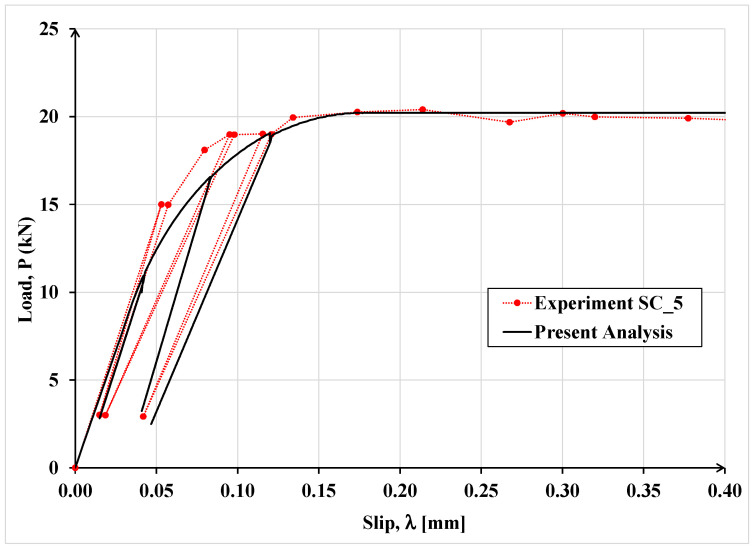
Comparison of experimental and analytical load-slip curves for specimen SC_5 subjected to cyclic load.

**Figure 19 materials-15-08690-f019:**
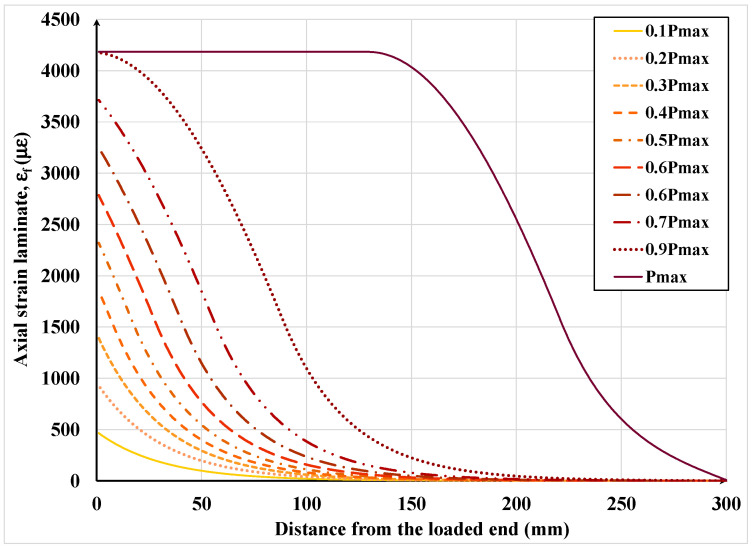
Development of steel textile axial strain along the bonded length under increasing load.

**Figure 20 materials-15-08690-f020:**
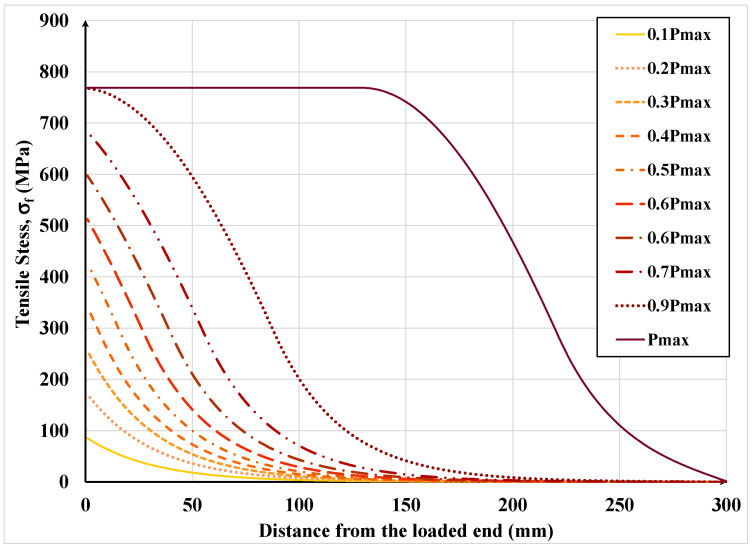
Development of steel textile tensile stress along its bonded length.

**Figure 21 materials-15-08690-f021:**
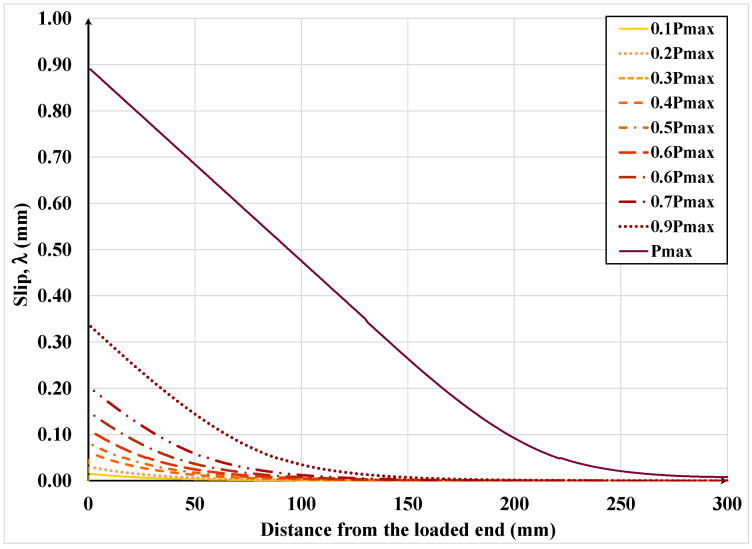
Slip variation along the bonded length under increasing load.

**Figure 22 materials-15-08690-f022:**
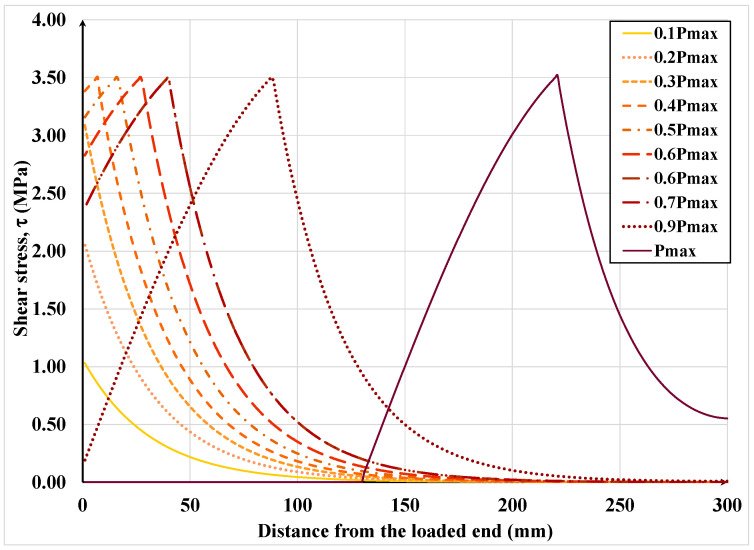
Interfacial shear stress variation along the bonded length under increasing load.

**Figure 23 materials-15-08690-f023:**
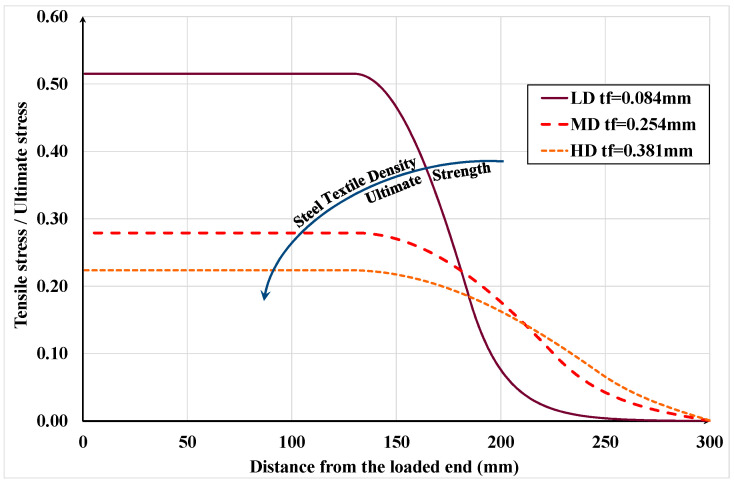
Normal strain behavior for different steel textile density.

**Figure 24 materials-15-08690-f024:**
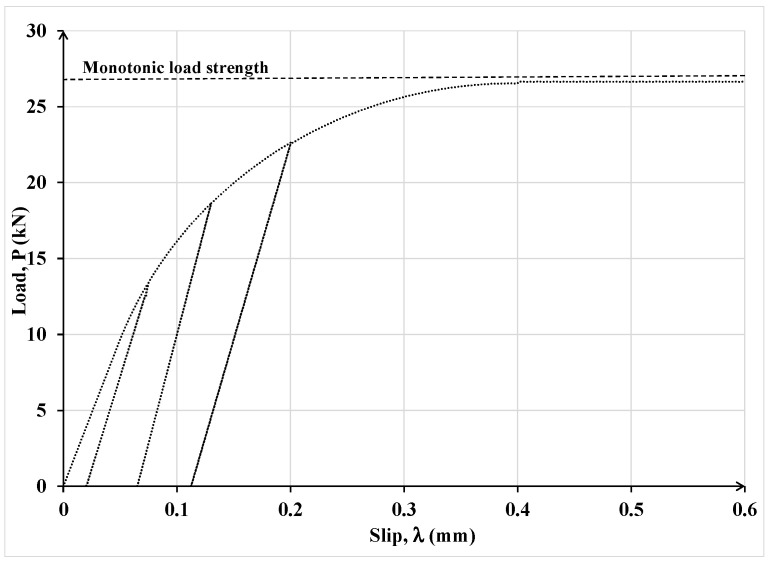
Load-slip curve under cyclic load.

**Figure 25 materials-15-08690-f025:**
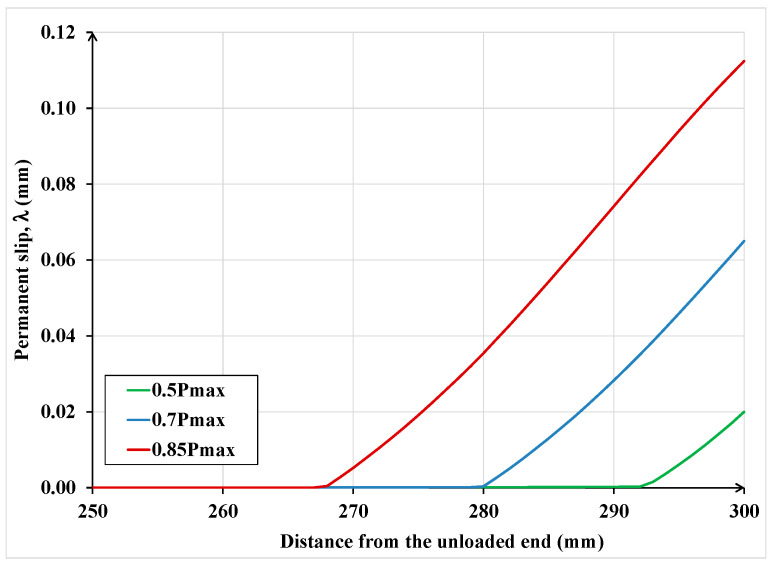
Permanent slip distribution along the bonded length under different load cycles.

**Figure 26 materials-15-08690-f026:**
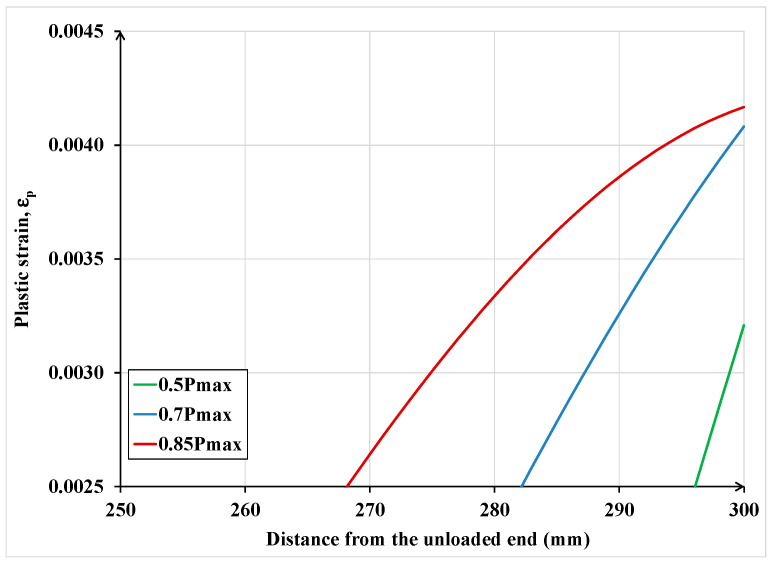
Plastic strains distribution along the bonded length.

**Table 1 materials-15-08690-t001:** Checking FE solution convergence through mesh refinement.

Trial	Element Size	Concrete	FRP Sheets	Force
	[mm]	Number of Finite Elements		[kN]
1	25	6000	600	28.210
2	20	7500	750	27.124
3	15	100,000	1000	26.900
4	10	150,000	1500	26.794
5	5	300,000	3000	26.785

**Table 2 materials-15-08690-t002:** (**a**) Mechanical properties of Concrete and FRP sheets. (**b**) Mechanical properties of the adhesive.

(**a**)			
		**Unit**	**Value**
Concrete Young’s Modulus	Ec	MPa	34,000
FRP Young’s Modulus	Ef	MPa	216,000
(**b**)			
		**Unit**	**Value**
Elastic Stiffness	k_S_	N/mm	100.0
Tensile Strength	**τ** _II_	MPa	5.0
Fracture Energy	G_II_	MPa∙mm	1.0
Ultimate displacement	s_u_	mm	0.4

**Table 3 materials-15-08690-t003:** Geometrical and material properties of specimens analyzed.

Property	Unit	Specimen		
		Case (a) [38]	Case (b) [1]	Case (c) [39]
Specimen width, $b_{c}$	mm	100	200	120
Specimen height, h	mm	100	150	120
Specimen strength, $f_{cs}$	MPa	15 -	22.5 -	4.4 14.8
Specimen elastic modulus, $E_{cs}$	GPa	25 -	28 -	7.8 19.5
FRP laminate thickness, $t_{f}$	mm	1.016 - -	0.084 0.254 0.381	0.254 - -
FRP laminate length, $L_{f}$	mm	203.2	300	200
FRP laminate width, $b_{f}$	mm	25.4	100	50
FRP laminate elastic modulus	GPa	110.4	-	-
Steel textile yield stress, $f_{y}$	MPa	-	2410	2410
Steel textile ultimate stress, $f_{u}$	MPa	-	3191	3191
Steel yield strain, $\varepsilon_{y}$	-	-	0.013	0.013
Steel strain corresponding to its ultimate strength, $\varepsilon_{su}$	-	-	0.021	0.021
Maximum shear stress, $\tau_{\max}$	MPa	5.75	2.6	3.5
Slip corresponding to $\tau_{\max}$, $\lambda_{0}$	mm	0.05	0.05	0.04
Ultimate slip, $\lambda_{u}$	mm	0.32	0.40	0.13

## Data Availability

The data required to reproduce these findings are all reported in the paper.

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
