# Peer review of "Nonlinear Analytical Procedure for Predicting Debonding of Laminate from Substrate Subjected to Monotonic or Cyclic Load"

_materials, 2022, doi:10.3390/ma15238690_

Round 1
Reviewer 2 Report
1) Why is only FRP material selected to be discussed and examined experimentally, even though in the background section there are two types of materials (SRP and FRP materials) that are discussed?
2) Describe the experiment's procedures and methods, including detailed drawings of specimens.
3). In section 5, there is a discussion of CFRP material, whereas in the previous section there was no explanation. Please add an explanation.
4). In the experiment, there are several loading conditions. Which type of loading conditions is suitable for the material? Does it depend on the type of material in each loading condition?
5). In which parts of the structure in the industry, this type of joint can be applied?
Reviewer 3 Report
The manuscript is of scientific interest and provides interesting results, however some corrections are required.
All the references should be placed in square brackets or replaced by superscripts.
Fig. 1. It is required to explain, what is the difference between (a), (b) and (c) parts in figure caption.
I guess, it would be better to rotate fig. 1(c) to make the same loading directions in fig 1 (b) and (c).
It should be explained within the text, what are τ(x), Nf, uf, uc before fig. 2.
The same for σf.
Line 166. refs. 41-5. It is unclear. Authors should check this place one more time.
There are some misprints within the text. Authors should carefully check all the text again.
Table 2b. Whether it is convenient to use fracture energy in KJ/m2 while elastic stiffness is in N/mm and ultimate displacement in mm? Wouldn't it be better to transform m to mm or vice versa?
In fig. 10 differences between FEM and proposed analytical model are clearly seen in unloading part. Authors should describe, whether FEM or proposed model better describe behavior during unloading cycle.
Table 3. FRP laminate thickness. Are the provided values of significant accuracy? Is it possible that 1.016 mm = 1.0 mm; 0.084 mm = 0.1 mm and 0.254 mm = 0.3 mm?
Authors should provide confidence intervals for obtained numerical data.
Is it possible to apply proposed model for simulation of behavior some other materials, not only FRP/SRP laminates?
Authors did not fill the "Acknowledgments" part. I guess, if there are no aclnowledgments, authors should remove this point.
Round 2
Reviewer 1 Report
The revised manuscript is recommended for publication.
Reviewer 3 Report
The manuscript was improved and could be published.